# Manual Annotation of Time in Bed Using Free-Living Recordings of Accelerometry Data

**DOI:** 10.3390/s21248442

**Published:** 2021-12-17

**Authors:** Esben Lykke Skovgaard, Jesper Pedersen, Niels Christian Møller, Anders Grøntved, Jan Christian Brønd

**Affiliations:** Centre of Research in Childhood Health, Research Unit for Exercise Epidemiology, Department of Sports Science and Clinical Biomechanics, University of Southern Denmark, 5230 Odense, Denmark; jespedersen@health.sdu.dk (J.P.); ncmoller@health.sdu.dk (N.C.M.); agroentved@health.sdu.dk (A.G.); jbrond@health.sdu.dk (J.C.B.)

**Keywords:** sleep, accelerometry, labeling, machine learning, physical activity, human behavior, circadian rhythms, classification, sleep/wake cycles, annotation, wearable sensors

## Abstract

With the emergence of machine learning for the classification of sleep and other human behaviors from accelerometer data, the need for correctly annotated data is higher than ever. We present and evaluate a novel method for the manual annotation of in-bed periods in accelerometer data using the open-source software Audacity^®^, and we compare the method to the EEG-based sleep monitoring device Zmachine^®^ Insight+ and self-reported sleep diaries. For evaluating the manual annotation method, we calculated the inter- and intra-rater agreement and agreement with Zmachine and sleep diaries using interclass correlation coefficients and Bland–Altman analysis. Our results showed excellent inter- and intra-rater agreement and excellent agreement with Zmachine and sleep diaries. The Bland–Altman limits of agreement were generally around ±30 min for the comparison between the manual annotation and the Zmachine timestamps for the in-bed period. Moreover, the mean bias was minuscule. We conclude that the manual annotation method presented is a viable option for annotating in-bed periods in accelerometer data, which will further qualify datasets without labeling or sleep records.

## 1. Introduction

Utilizing machine learning with the identification of sleep, physical activity behavior, or non-wear from accelerometry data provides the ability to model very complex and non-linear relationships, which is not possible with more simple statistical methods, like multiple linear or logistic regression [1]. However, the use of supervised machine learning algorithms demands large amounts of accurate annotated data to provide sufficient accuracy and to ensure generalizability [2]. Sleep is increasingly recognized as a critical component of the healthy development of children and overall health [3,4,5], and healthy sleep is generally defined by adequate duration, appropriate timing, good quality, and the absence of sleep disturbances or disorders [6].

Nevertheless, only scarce efforts to assess sleep measures from accelerometer data have been successfully approached with advanced machine learning techniques [7] providing researchers with inexpensive and minimally invasive methods. Valid objective measures using accurate automated scoring of sleep and wake time are important to provide valuable insights into the circadian rhythms on an individual and population-wide basis. The gold standard for objective sleep assessment is polysomnography (PSG), which is based on the continuous recording of electroencephalographic (EEG), electromyographic (EMG), and electrooculographic (EOG) activity via surface electrodes.

The assessment of sleep from PSG provides a detailed description of individuals’ sleep architecture through the identification of various sleep stages in addition to the more general sleep outcomes, like sleep duration and timing [8]. The PSG method is costly and burdensome in terms of technician support for sensor application/removal, overnight monitoring (for in-lab PSG), and manual record scoring, in addition to being intrusive for the patient due to the necessity of wearing multiple sensors on the scalp and face throughout the night.

Recent studies have attempted to identify PSG-assessed sleep-wake classification with wrist acceleration using machine-learning techniques [7,9,10]. One study by Sundararajan et al. [9] attempted this using a random forest machine learning algorithm. The results from the study revealed an F1 score of 73.9% as an estimate of overall accuracy for the identification of sleep-wake stages. Moreover, the study showed a large false discovery rate (i.e., the true wake time was predicted as sleep by the algorithm) for the identification of sleep.

This may primarily be a result of the intrinsic limitations of a wrist-worn accelerometer incorrectly classifying quiet wakefulness as sleep and, secondly, may be explained by the number of participants and the use of single-night PSG-recordings restricting information on inter- and intraindividual variation. Thus, the poor false discovery rate could potentially be balanced out by increasing the number of subjects in the study and/or by increasing the number of consecutive days of recordings for more information on the variation in the movement behavior of the subjects during sleep hours.

Accelerometry provides researchers with an inexpensive and minimally invasive method, which has the potential to play an important role in the assessment of sleep duration and timing characterization since it is more practical and suitable than PSG for prolonged recordings (i.e., multiple consecutive days) in non-laboratory settings [11]. However, the limitations of accelerometry must be acknowledged, which is also emphasized by the poor results for the identification of sleep vs. wakefulness with wrist acceleration presented with the study by Sundararajan et al.

Thus, accelerometry does impose certain limitations as a methodology to identifying subject’s sleep behavior and developing new algorithms should not focus on sleep staging but rather the sleep timing (bedtime/wake-up time) and specifically the sleep/wake state of the subject. Moreover, the accurate identification of the sleep-wake state from accelerometry is most optimally approached with accelerometry recordings covering at least 7–10 days of measurement for each subject to ensure the appropriate day-to-day variation and movement behavior during sleep hours.

Developing supervised machine learning algorithms to identify the sleep/wake cycle from multiple days of accelerometry recordings of individuals requires the annotation of the data to identify time in bed and specifically when the participants go to bed and when they get out of bed. Although there is an obvious distinction between time in bed and actual sleep time, accelerometry as a surrogate measure of sleep is widely used in the literature [12,13,14,15] due to the many practical advantages of using accelerometry compared to more intricate methodologies for the detection of sleep.

The time in bed annotation could be established from individual sleep diaries, EEG-based recordings [16], systems for the recording of tracheal sounds [17,18], etc.; however, such additional data are not recorded in conjunction with accelerometry within many studies, which leaves a substantial data resource ready for enrichment. If individual time in bed periods can be accurately annotated without additional data for correct labeling, it would provide the option to use the accelerometer data to facilitate the improvement of existing algorithms or the development of new supervised machine learning algorithms. Currently, there are no accurate or easy-to-use methods for the manual annotation of time in bed from accelerometry.

The aims of the present study were to (1) describe a method for the manual annotation of subjects’ individual bedtime and time out of bed with raw unprocessed accelerometry, (2) evaluate the accuracy of the manual annotation to predict time in bed/out of bed obtained using a single channel EEG-based sleep staging system and a sleep diary, and (3) to evaluate the inter- and intra-rater reliability of annotations.

## 2. Materials and Methods

### 2.1. Study Population 

Data for the current study originates from the SCREENS pilot trial (www.clinicaltrials.gov (accessed on 15 May 2020), NCT03788525), which is a two-arm parallel-group cluster-randomized trial with two intervention groups and no control group [19,20]. Data were collected between October 2018 and March 2019.

The collection of data was reported to the local data protection department SDU RIO (ID: 10.391) in agreement with the rules of the Danish Data Protection Agency.

Families in the municipality of Middelfart in Denmark were invited to participate if they had at least one child aged 6–10 years residing in the household (*n* = 1686). Based on survey responses, families were eligible to participate if the contacted parent’s total screen media use was above the median amount (2.7 h/day) based on all respondents (*n* = 394) and if all children in the household were older than 3.9 years.

The latter was to avoid potential disturbances of sleep measurement due to an infant or toddler’s pattern of nocturnal awakening. For further details on inclusion and exclusion criteria see Pedersen et al. [21] In total, data from 14 children and 19 adults were included in the present study. The included participants were not instructed to change their sleep and bedtime behavior as a part of the interventions. The napping behavior, if any, of the participants was irrelevant to the current study as we focused on their nightly sleep time monitored by the EEG-based sleep staging system.

### 2.2. Actigraphy

Both adults and children underwent 24-h accelerometry recording using two Axivity AX3 (Axivity Ltd., Newcastle upon Tyne, UK) triaxial accelerometers. The Axivity AX3 is a small (dimensions: 23 mm × 32.5 mm × 7.6 mm) weighing only 11 g. Sensitivity was set to ±8 g and the sampling frequency to 50 Hz. The accelerometers were worn at two anatomical locations; one fixated to the body in a pocket attached to a belt worn around the waist, where the sensor was placed on the right hip with the USB connector facing away from the right side of the body.

A second belt was worn around the right thigh midway between the hip and the knee, where the accelerometer was placed in a pocket with the USB connector facing away from the body. The devices were worn for 1 week (seven consecutive days) at baseline and at follow-up, which corresponds to the recommended number of days required to reliably estimate habitual physical activity [22].

### 2.3. Zmachine^®^ Insight+

Concurrent with the accelerometer recordings, both adults and children underwent sleep assessments for 3–4 nights at baseline and 3 nights at follow-up using the Zmachine^®^ (ZM) Insight+ model DT-200 (General Sleep Corporation, Cleveland, OH, USA), Firmware version 5.1.0) concurrently with the actigraphy recording. The device measures sleep by single-channel EEG from the differential mastoid (A1–A2) EEG location on a 30-s epoch basis. The sleep apparatus is developed for use in a free-living setting for objective measurement of sleep, including measurement of sleep duration and sleep stage classification, as well as computation of sleep-specific quantities, e.g., latency to the respective sleep stages.

The algorithm in ZM has been compared to polysomnography (PSG) in adults with and without chronic sleep issues within a laboratory setting [23,24], and we found that ZM can be feasibly applied to children and adults for multiple days of measurements in free-living [12]. The ZM device demonstrated a high degree of validity for detecting sleep versus awake with a sensitivity, specificity, positive predictive value, and negative predictive values of 95.5%, 92.5%, 98%, and 84.2%, respectively [24].

Three electrodes (Ambu A/S, Ballerup, Denmark, type: N-00-S/25) are mounted on the mastoids (signal) and the back of the neck (ground). Thirty minutes before the participants plan to go to bed to sleep, the skin areas are cleansed with an alcohol swab, and then electrodes are attached to the skin. An EEG cable connects the three electrodes to the ZM device, where a sensor check is performed to detect whether one or more electrodes are not mounted correctly. If there are sensor problems, these are solved swiftly by a simple change of said electrodes. Participants also reported their bedtimes and time of awakening each day using a prospective daily diary. Parents reported the bedtimes and times of awakening on behalf of their child.

### 2.4. Audacity

Audacity is a free software tool that was developed for audio editing. The software project was originally started by Dominic Mazzoni and Roger Dannenberg in the fall of 1999 as part of a research project at Carnegie Mellon University. Audacity was initially released as an open-source audio editor in May 2000. Since then, the software has been community-developed adding hundreds of features, providing complete support for professional-quality 24-bit and 32-bit audio, a complete Manual and support for many different languages, and millions of copies have been distributed. Today, Audacity is being maintained by a team of volunteers located in many different countries. Audacity is distributed under the terms of the GNU General Public License. Everyone is free to use this application for any personal, educational, or commercial purpose.

In the context of analyzing accelerometer data, Audacity is especially useful as it enables the researchers to effortlessly inspect high-resolution raw accelerometer data with a high degree of precision. It is possible to rapidly zoom in to inspect portions of the accelerometer recording in detail (e.g., to inspect certain behavior around bedtime) or zoom out to get an overview (e.g., a whole week). Moreover, Audacity provides a high-resolution labeling function that can be used for the annotation of the accelerometry data. All labels created can be stored in a separate file and subsequently used in the ML algorithm. The ability to manually inspect high-resolution raw accelerometer data at the level of detail provided by Audacity is, to the knowledge of the authors of the current study, unprecedented in other software.

In the Audacity software, it is possible to combine more than 100 channels of data, which provides the ability to combine different signal features derived from the acceleration. Adding multiple signal features together provides an interesting option that might facilitate the visual interpretation and classification of the underlying behavior. However, adding too many signal features could have negative consequences for the accurate identification of the behavior of interest. We combined a total of seven independent signal features. The classification of “lying” within the first feature is derived as follows: the inclination of the hip accelerometer exceeds 65 degrees, and the thigh accelerometer is concurrently classified as “sitting” by the activity type classification algorithm by Skotte et al. [25]. The remaining signal features, excluding “time”, were selected from algorithm by Skotte et al. directly. All included features are summarized and described in Table 1. The accelerometry signal features are described with respect to the longitudinal axis of the body. The features generated from the accelerometry data is processed using a window length of two seconds (60 samples) and a 50% overlap (30 samples), providing a resolution of one second. The algorithm by Skotte et al. and the algorithm producing the first feature are solely based on the inclination(s) of the accelerometer(s) and, as such, can be used not only to assess the time in bed but also the posture of the participants. Therefore, this is not a precise indicator of the to-bed/out-of-bed timestamps.

Figure 1 and Figure 2 shows examples of the visual Audacity interface with all seven signal features as listed in Table 1. Figure 1 is a seven-day overview, and Figure 2 presents a zoomed view of approximately 24 h and with a single annotated night.

### 2.5. Manual Annotation by the Raters 

The three raters were all researchers who had prior experience working with accelerometer data, and thus, had some understanding and knowledge on how to interpret the different data channels available. The raters labeled each wav-file independently of each other with in-bed and out-of-bed timestamps and exported the corresponding labels as text files. Each file was labeled twice (round 1 and round 2) for test–retest purposes. The raters were at no time aware of previous annotations made by themselves or by the other raters.

### 2.6. Ground Truth

The ZM ground truth labels of the time in-bed and out-of-bed events were derived from the sleep staging data of the ZM as the first and last non-sensor-problem event for the night. If the ZM reported the beginning or the end of the recording as having sensor problems, the corresponding night was discarded from further analysis. Sensor problems most commonly occur due to poor attachment of the electrodes. All subjects were instructed to attach the ZM and turn it on at the exact time when the participants went to bed and to remove upon awakening. The timestamps of these events were used as the ground truth values.

### 2.7. Statistical Analysis

All statistical analyses were performed using R statistical (R Core Team, Vienna, Austria) software version 4.0.2 (22 June 2020), RStudio (RStudio Inc., Boston, MA, USA) version 1.1.456. Descriptive characteristics were computed using medians and interquartile ranges for continuous variables and proportions for categorical variables. Characteristics are presented separately for children and adults.

Agreement analyses were performed using intraclass correlation coefficient (ICC) and Bland–Altman analysis. Furthermore, to illustrate the overall agreement and symmetry of methods, probability density distribution plots are shown. The ICC is an index that, contrary to Pearson correlation, assesses not only how well correlated the two techniques are but also if they are equal. An ICC < 0.5 indicates poor agreement, 0.5 < ICC > 0.75 indicates moderate agreement, 0.75 < ICC > 0.9 indicates good agreement, and ICC > 0.90 indicates excellent agreement [26].

In the current study, interpretations of the ICCs are based on the corresponding 95% confidence intervals in accordance with guidelines [26]. Bland–Altman analyses allow examining the degree of agreement between two measurement techniques [27]. The mean of the differences between two techniques (representing the mean bias) and limits of agreement (which are defined as a deviation from the mean superior to two standard deviations) are calculated. A positive bias/mean difference indicates an underestimation (earlier) of the to-bed or out-of-bed timestamp, while a negative difference indicates an overestimation (later) compared to ZM.

## 3. Results

Descriptive characteristics of the included subjects of the current study are reported in Table 2.

### 3.1. Intraclass Correlation Coefficient Analyses

The ICC analyses highlighted an excellent agreement between ZM and manual in-bed annotation for time to bed and time out of bed across both rounds 1 and 2 and at the baseline and follow-up with the lower limits of the confidence intervals all above 0.9 (see Table 3).

Excellent agreement was also observed between self-report and ZM for both baseline data and follow-up data, which is indicated by the lower limit of the 95% confidence interval has values no less than 0.94 (see Table 4).

The ICCs of the agreement between the three manual raters’ ability to annotate the to bed and out of bed timestamps showed good to excellent agreement as seen by the lower limits of the 95% confidence intervals no less than 0.88. A slight tendency of difference of the ICCs can be seen when comparing the to-bed to the out-of-bed timestamps (see Table 5).

The ICCs for the test–retest reliability showed good to excellent agreement for each rater between rounds 1 and 2 for both baseline data and follow-up data (see Table 6). This is seen by the lower limits of the 95% confidence intervals values of no less than 0.86. Although the ICCs are similar, it seems that raters 1 and 3 showed lower agreement when annotating the baseline to-bed timestamp compared to the later annotations, whereas the ICC scores of rater 2 did not elicit this behavior.

### 3.2. Bland–Altman Analyses

The bias and upper and lower limits of agreement with corresponding confidence intervals for the comparison of the manual annotation and self-report in relation to ZM are presented in Table 7. Biases for the manual annotation compared to ZM is in the range of -6 min to 5 min, while self-report produced slightly lower biases in comparison to ZM. Generally, the limits of agreement seem to be of the same magnitude regardless of the method comparison.

### 3.3. Density Plots

The probability density distribution for the difference between the to-bed and out-of-bed scoring for the manual annotation and self-report compared to ZM is shown in Figure 3. These plots function as a visual representation of the bias and spread around zero of the manual annotations and self-report in comparison to ZM as seen previously [10].

## 4. Discussion

This is the first study to describe a method for the manual annotation of in-bed periods with accelerometry data and to evaluate the accuracy of the method with multiple raters and compared to sleep assessed with EEG -based methodology Zmachine Insight+^®^ and self-reported sleep as reference methodologies. When all interpretations of the ICC analyses were based on the lower limit of the 95% confidence interval, our results showed (1) good-to-excellent interrater reliability, (2) the test–retest reliability (or intra-rater reliability) showed good to excellent agreement for all three raters between their first and second round of in-bed annotations, (3) compared to ZM, the average of the manual in-bed annotation method for all three raters showed agreements ranging from good to excellent, and (4) the agreement between the self-reported in-bed timestamps and ZM were good to excellent. Furthermore, the Bland–Altman analysis revealed that the mean bias of the manual annotation and self-report compared to ZM was within ±6 min with LOA no larger than a span of ±45 min. Finally, the probability density distribution plots of the differences between the in-bed estimates of the manual raters and self-report compared to ZM were comparable in terms of the symmetry, spread around zero, and positioning of outliers.

The excellent performance of the prospective sleep diaries in the current study may be explained by the synchronized use of ZM and the sleep diaries. Thus, having the subject manually initiate and end the ZM recording every morning and night will make it easier for the subject to recall the time going to bed and time out of bed, thus, avoiding much of the usual discrepancy between objectively and subjectively measured sleep duration [28]. We would not expect to see such good agreement between the manual annotation and sleep diaries as well as ZM and sleep diaries if the participants wre instructed to exclusively log sleep using subjective measurements without protocol disturbances as anchor time points.

When compared to ZM, we found that the manual annotation of the in-bed period deviated more when estimating the going to bed timestamp. This could be caused by the difficulty the raters had in discriminating between inactive behaviors before bedtime and actual time in bed. However, these discrepancies may be minimized by further training of the rater’s ability to distinguish between inactive behaviors; however, this poses the most important limitation to the manual annotation method. Nevertheless, the accuracy obtained in the present study is reassuring as it is achieved based on little preliminary formal training or briefing of the raters involved. In that sense, most of the work when determining the in-bed period is self-explanatory when provided with the information, given by the signal features selected in the present study in Audacity. This is further supported by the excellent ICC between the three manual raters. However, there appears to be a slight learning curve as the LOAs are consistently narrower during round 2 of the manual scoring compared to round 1. This is also evident in the density plots, which display a greater spread around zero during round 1 compared to round 2. This suggests that more than two rounds of manual scoring may homogenize the results further or that the raters may benefit from revisiting the annotations from the beginning of the first round of annotations. Alternatively, a form of training may be profitable before the actual annotation takes place. Further research is warranted to investigate methods to optimize the homogenization of the manual annotations. Nevertheless, the evidence suggests that supervised machine learning, given a large amount of labeled data, is resistant to label noise [29], which means that the tradeoff for accuracy in favor of the sheer volume of labeled data may be preferable. This further advocates for the use of the manual annotation methodology in data sets with no self-reported sleep or other measures of interest without labels.

There are other labeling tools for annotating time series data (e.g., Label Studio (Heartex Organization, San Francisco, CA, USA) [30] or Visplore (Visplore GmbH, Vienna, Austria) [31]); however, we found that Audacity was well suited for this specific task. Label Studio, for instance, may have difficulties handling week-long accelerometer data with 100 M+ entries, and Audacity is perfectly suited with its ability to seamlessly handle and navigate very large data structures. Furthermore, our feature selection was based on domain knowledge with the purpose of providing the right combination of features in limited number to avoid overflowing the raters with redundant information. This methodology can be extended to other behaviors, e.g., walking, which would likely require a different set of features. Although we do not provide clear-cut guidelines for the process of annotating the data, the raters in the present study were able to gain the right insights. Furthermore, the labeling of data is a step that inherently requires common knowledge in human behavior, and if the labels can sufficiently be described based on formal rules, one can question whether the training of an AI model is necessary at all. Nevertheless, we suggest that further research investigating which features are the most important for successful annotations and, likewise, examine the effect of other sets of features that might provide important knowledge that could further facilitate the use of manual annotation of accelerometry time series data.

To date, most studies that have investigated the validity of actigraphy and self-report compared to PSG or EEG-based methodologies, have routinely evaluated sleep parameters, such as the total sleep time, wake after sleep onset, sleep latency, and sleep efficiency. These measures are often an aggregate measure that includes sleep onset and wake onset, which would be comparable to the to-bed and out-of-bed timestamps in the current study. Moreover, the precision of these time points is not evaluated and, thus, is difficult to compare to the measurements of the current study. The novel methodology for annotating the time in bed in the present study, however, provides ICC values on par with or better than previous studies comparing actigraphy sleep parameters to PSG [7,32]. Furthermore, one study presented mean absolute errors of 39.9 min and 29.9 min for the sleep onset time and waking up time, respectively, and 95% limits of agreement above ±3 h for sleep duration when comparing an algorithm to PSG [10]. Furthermore, the ability to precisely estimate the timestamp of certain events compared to durations of specific behaviors leaves less room for error in the effort to obtain good agreements. Additionally, our manual methodology performs strongly across age and gender as the included subjects in the present study consist of both children and adults of both genders. This suggests that our manual annotation method is accurate irrespective of the inclusion of different developmental age groups and genders and their specific behavior and that it may be a more precise tool for estimating exact time points compared to present state automated methodologies. Traditionally, the accuracy of the assessment of sleep parameters is highly dependent on the target population, and thus we view the current results with plausible high generalizability to populations of normal sleepers.

Identifying periods of sleep (rather than simply lying) is an important component of a 24 h behavior profile, and many studies examining sleep detection based on conventional accelerometers involve asking the participant to record their time in bed, sleep onset, and waking up time [33,34,35]. The use of self-reported measures of sleep may be replaced by annotating in-bed manually, thereby, lowering the participant burden and avoiding the inevitable recall bias associated with self-reported measures. Therefore, an important application of the manual annotation methodology using Audacity is that it can, with no difficulty, be employed on free-living data. Moreover, the application of our proposed methodology is manifold. Other suitable use cases could be the annotation of non-wear time, manual clock synchronization of several different devices, examining the validity of raw data, and more. Additionally, it is not limited to actigraphy data but can be utilized on a wide variety of multi-channel data for an increased overview, including orientation (gyroscopic data), temperature, battery voltage, and light as examples. Finally, Audacity provides a fluid workflow even with very large multi-channel data files with the ability to swiftly zoom to every resolution needed and scroll through time with no lag and add labels that advocate the use of Audacity as a standard tool for researchers working with raw data and machine learning. For these purposes, the implementation of the Audacity-methodology on raw accelerometer data may help drive the development of future human behavior research.

Although the work of changing from raw sensor data to operational predictive models using labelled data has been the standard method for years, no previous studies have proposed a methodology that enables researchers to make optimal use of their available accelerometry data. We show that with a few well-selected features, the annotation of sleep is comparable to EEG-based sleep classification hardware. However, it is important to note that we do not currently recommend that our proposed methodology of manually annotating in-bed time is to be used as a replacement of other more well-established techniques for estimating sleep, e.g., EEG- or tracheal-sound-based options in ongoing studies. Though, it may serve as a post-hoc procedure to enrich already collected data with a measure of sleep.

The strengths of this study include the continuous data collection of both accelerometry, sleep diary, and ZM in the home of the participants during multiple consecutive days of recordings making it high-quality free-living data. The limitations include the limited rater generalizability as the three manual raters were fixed and not randomly chosen from a larger population of eligible raters to accommodate different characteristics. However, due to the scarce pre-briefing instructions on how to label the raw data, we suggest that this methodology is generalizable to at least other researchers working with accelerometer data.

A natural next step would be to develop and validate a procedure similar to what is available with sleep annotation with EEG (AASM Scoring Manual [36]) and propose guidelines to make the manual annotation methodology accessible to persons with limited experience within the field of accelerometry. To record true free-living behavior using participant-mounted devices is difficult and wearing the ZM during sleep may affect the behavior of the participants and, thus, poses as a limitation of the study.

Additionally, although the criterion measure in the current study is validated against PSG, it would have been more optimal to use PSG as a criterion measure. Finally, we did not incorporate napping behavior in the current study as we focused on sleep in relation to circadian rhythms. Further research is needed to validate the manual methodology for use in detecting napping behavior.

## 5. Conclusions

In conclusion, our results show that the manual annotation of the in-bed period from thigh- and hip-worn accelerometer data using Audacity demonstrated good agreement with a minimal mean bias and acceptable limits of agreement for the time to bed and out of bed when compared to the same estimates assessed with the use of an objective EEG-based sleep device and prospective sleep dairies. Furthermore, the manual annotation was highly reliable with excellent inter- and intra-rater agreement and has an accuracy with the EEG-based assessment similar to the sleep diaries.

The study shows that the manual annotation can be used on already collected raw data when not accompanied by sleep records. This will facilitate the additional use of free-living data resources and, thus, could increase the amount of available training data when employing data-demanding machine learning algorithms. This has the potential to improve the generalizability of these algorithms in assessing human behavior from objective recordings.

## Figures and Tables

**Figure 1 sensors-21-08442-f001:**
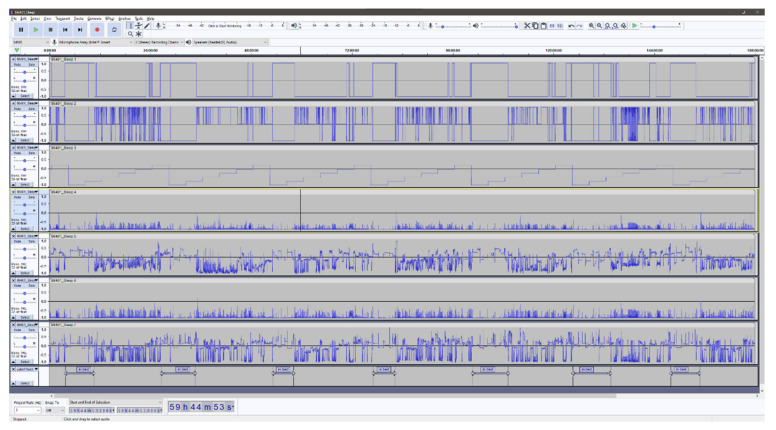
Screenshot of the Audacity interface showing the seven horizontal panels representing the included signal features. See Table 1 for a detailed description of the features.

**Figure 2 sensors-21-08442-f002:**
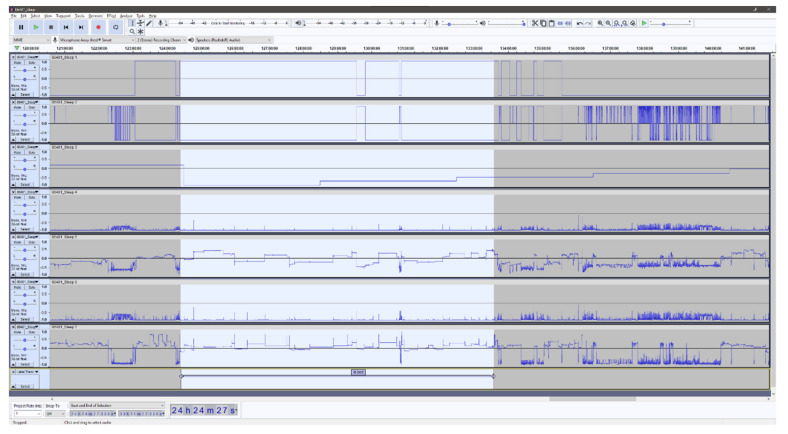
Screenshot of the Audacity interface when zoomed in on a single night for the labeling of the in-bed period. The seven horizontal panels represent the included signal features. See Table 1 for a detailed description of features.

**Figure 3 sensors-21-08442-f003:**
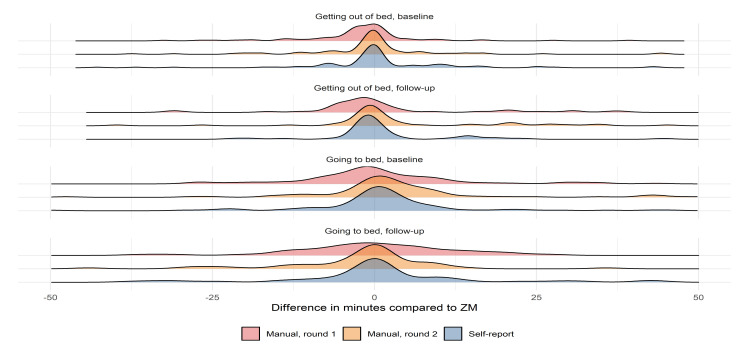
Probability density distributions for differences between manual in-bed annotations and self-report compared to ZM.

**Table 1 sensors-21-08442-t001:** Signal features for the detection of in-bed periods in Audacity.

Name	Description	Values	Visual Interpretation
Lying	The classification of lying based on the thigh and back	1: lying −1: not lying	Lying position
Activity	The classification of activity type	1: Standing, moving, or walking 0: Sitting −1: Other activity	This feature will guide the rater to identify periods of activity prior to correct bedtime
Time	Time categorized into four-hour windows	−1: 00:00–04:00 and so on throughout the 24 h cycle	Time of day
Thigh-SDacc	Standard deviation of the acceleration on the longitudinal axis of the thigh	−1: No movement	Proportion of leg movement
Thigh-Inclination	Inclination angle of the thigh device in relation to the longitudinal axis of the thigh	The −1 to 1 range represents the −180 to 180 degrees inclination angle	Inclination angle of the thigh
Hip-SDacc	Standard deviation of the hip acceleration on the longitudinal axis of the torso	−1: No movement	Proportion of whole-body movement
Hip-Inclination	Inclination of the hip device in relation to the longitudinal axis of the torso	The −1 to 1 range represents the −180 to 180 degrees inclination angle	Inclination angle of the body/hip

**Table 2 sensors-21-08442-t002:** Descriptive characteristics of the study participants.

Population (*N* = 33)	
Children	
*N*	14
Gender (% female)	28.6
Age (years)	9 (7–10)
Adults	
*N*	19
Gender (% female)	57.9
Age (years)	42 (39–46)
ISCED	
0–3 (%)	36.8
4–6 (%)	47.4
7–8 (%)	15.8
*ISCED, International Standard Classification of Education*	

**Table 3 sensors-21-08442-t003:** Intraclass correlation coefficients between ZM and the average of the manual annotations between the three raters.

	Baseline (*n* = 94 Nights)		Follow-Up (*n* = 54 Nights)	
	Round 1 ICC (95% CI)	Round 2 ICC (95% CI)	Round 1 ICC (95% CI)	Round 2 ICC (95% CI)
To bed	0.98 (0.98; 0.99)	0.98 (0.96; 0.98)	0.96 (0.94; 0.98)	0.95 (0.92; 0.97)
Out of bed	0.98 (0.97; 0.99)	0.98 (0.96; 0.98)	0.98 (0.97; 0.99)	0.97 (0.95; 0.98)

Round 1 and round 2 refers to the first and second round of annotation.

**Table 4 sensors-21-08442-t004:** Intraclass correlation coefficients between self-report and ZM.

	Baseline (*n* = 94 Nights)	Follow-Up (*n* = 54 Nights)
	ICC (95% CI)	ICC (95% CI)
To bed	0.98 (0.98; 0.99)	0.96 (0.94; 0.98)
Out of bed	0.98 (0.97; 0.99)	0.98 (0.96; 0.99)

**Table 5 sensors-21-08442-t005:** Intraclass correlation coefficients between manual raters.

	Baseline (*n* = 110 Nights)		Follow-Up (*n* = 62 Nights)	
	Round 1 ICC (95% CI)	Round 2 ICC (95% CI)	Round 1 ICC (95% CI)	Round 2 ICC (95% CI)
To bed	0.91 (0.88; 0.94)	0.92 (0.89; 0.94)	0.94 (0.9; 0.96)	0.97 (0.95; 0.98)
Out of bed	0.93 (0.9; 0.95)	0.97 (0.96; 0.98)	0.97 (0.96; 0.98)	0.98 (0.98; 0.99)

Round 1 and round 2 refers to the first and second round of annotation.

**Table 6 sensors-21-08442-t006:** Test–retest intraclass correlation coefficients between the first and second round of manual annotations.

	Baseline (*n* = 110 Nights)		Follow-Up (*n* = 62 Nights)	
	To Bed ICC (95% CI)	Out of Bed ICC (95% CI)	To Bed ICC (95% CI)	Out of Bed ICC (95% CI)
Rater 1	0.91 (0.87; 0.94)	0.98 (0.98; 0.99)	0.96 (0.94; 0.98)	1.00 (0.99; 1.00)
Rater 2	0.97 (0.96; 0.98)	0.91 (0.87; 0.94)	0.91 (0.86; 0.95)	0.99 (0.98; 0.99)
Rater 3	0.91 (0.87; 0.94)	0.96 (0.94; 0.97)	0.98 (0.97; 0.99)	0.98 (0.97; 0.99)

**Table 7 sensors-21-08442-t007:** Bland–Altman analysis of inter-method agreement between manual annotation and ZM as well as self-report and ZM. All estimates are in minutes.

Method	Bias (95% CI)	Upper LOA (95% CI)	Lower LOA (95% CI)
Baseline, to bed (*n* = 94)			
Manual, round 1	3.02 (−0.44; 6.47)	−30.04 (−35.96; −24.12)	36.07 (30.15; 42)
Manual, round 2	0.48 (−2.42; 3.39)	−27.3 (−32.28; −22.32)	28.27 (23.29; 33.24)
Self-report	1.23 (−1.57; 4.03)	−25.56 (−30.37; −20.76)	28.02 (23.21; 32.82)
Baseline, out of bed (*n* = 94)			
Manual, round 1	0.53 (−2.34; 3.4)	−26.9 (−31.82; −21.99)	27.96 (23.05; 32.88)
Manual, round 2	0.98 (−1.47; 3.43)	−22.49 (−26.7; −18.28)	24.45 (20.24; 28.66)
Self-report	−2.79 (−5.26; −0.32)	−26.45 (−30.69; −22.21)	20.87 (16.63; 25.11)
Follow-up, to bed (*n* = 54)			
Manual, round 1	−6.08 (−11.34; −0.83)	−43.81 (−52.84; −34.77)	31.64 (22.61; 40.67)
Manual, round 2	−0.4 (−5.3; 4.51)	−35.6 (−44.03; −27.17)	34.8 (26.37; 43.23)
Self-report	0.77 (−4.08; 5.62)	−34.06 (−42.4; −25.72)	35.59 (27.25; 43.93)
Follow-up, out of bed (*n* = 54)			
Manual, round 1	4.95 (0.65; 9.25)	−25.95 (−33.35; −18.55)	35.85 (28.45; 43.25)
Manual, round 2	2.57 (−0.76; 5.89)	−21.3 (−27.02; −15.59)	26.44 (20.72; 32.15)
Self-report	0.56 (−3.62; 4.74)	−29.45 (−36.64; −22.26)	30.57 (23.39; 37.76)

## Data Availability

The datasets generated and analyzed during the current study are not publicly available due to the general data protection regulations but will be shared on reasonable request using a safe platform by the corresponding author.

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
