# Peer review of "Manual Annotation of Time in Bed Using Free-Living Recordings of Accelerometry Data"

_sensors, 2021, doi:10.3390/s21248442_

Round 1
Reviewer 1 Report
Review for “Manual annotation of time in bed using free-living recordings of accelerometry data” by Esben Lykke, Jesper Pedersen, Niels Christian Møller, Anders Grøntved, and Jan Christian Brønd
Article summary:
This paper proposes a novel method of manually annotating time-in-bed recordings from accelerometer data using Audacity. The study is motivated by a need for a method to annotate data more easily with the aim of providing larger datasets for training supervised machine learning algorithms. The study recruited adults and children to participate in a sleep assessment for 3-4 nights (baseline) and 3 nights (follow-up) using a Zmachine Insight+ model DT-200 sensor as a ground truth metric and two accelerometers (one on the hip and one on the thigh). Each experiment was labeled by two different raters (three different raters were involved overall). Statistics on accuracy, intra, and inter-rater reliability were examined.
Review:
I would like to thank the authors for their updates and revisions. The changes have added significant clarification to the manuscript. However, the authors need to provide much more information to help readers understand the contribution of the work.
First, in examining Figures 1 and 2, to a naïve reader it appears that the labeling problem is almost trivial using the first feature, “lying” on its own. The only case where this does not appear to be a trivial problem is towards the end of the data shown in Figure 1, in which the lying label alternates between 1 and 0 a few times within one sleep period. However, this issue could be resolved with a minimum duration requirement. I think it would be exceptionally helpful for readers to understand the difficulty of the problem if examples were presented that showed data where the “lying” label may not correspond directly with sleep.
Related to this first point, if presented the data shown in Figures 1 and 2, it seems clear that any labeler, naïve or experienced, would be able to quickly identify segments based solely on the labels shown in the first two features (lying and activity classification). Labelers experienced with accelerometers would also quickly and easily see the decreased variance in the accelerometer signals during the periods in which the “lying” label is 1. This may explain why no labeling instructions are required, but if this is the case, then the labeling problem is trivial. Again, seeing examples of difficult labeling problems would be extremely beneficial for the readers to understand the contribution of this work.
Overall, the novelty of the work appears to be one of two things: 1) Audacity should be used to label accelerometer data. If this is the case, Audacity should be evaluated against the other tools used for labeling accelerometer data. EX: Skotte et. Al. used MATLAB software (Acti4). Are labelers unable to label the accelerometer data as accurately using this (or other) tool(s)? Is the problem simply that MATLAB (Acti4) takes more time to prepare for examining accelerometer data? Could MATLAB (Acti4) provide more insight than Audacity in some situations? What other tools are used by researchers working with accelerometer data? 2) However, if the novelty is the method of manually annotating accelerometer data to identify time-in-bed, then the process used for labeling is one of the most significant contributions of the work. Which features are most valuable to use when labeling? When challenging situations occur for labeling, what should a labeler do - what data matters? However, the authors have indicated that there was no instruction or process for the labeling of the data, so either the labeling process is trivial or a process was followed that needs to be articulated.
Also, Skotte et al is referenced and the algorithm is used in the paper (Line 183), but the paper by Skotte et al is not cited!
Reviewer 2 Report
1) It is still not clear why this system is better than EEG or tracheal sounds-based recording. Please make a clear paragraph in the discussion to make a distinction between them. There was no citation of EEG or tracheal sounds-based work as well.
Round 2
Reviewer 1 Report
Thank you for your replies and updates. The clarifications made in this round of revision significantly improve the clarity of the work presented. While the scope of this work is very narrow, the novelty and purpose are much more clear. The only concern that I have maintained across all reviews is the lack of a formal process or set of formal rules for annotation. However, the paragraph added to the discussion section describing the limitations addresses this to some extent.
Small formatting note before publication - Table 1 is currently split between two pages.
Reviewer 2 Report
No more comments.
This manuscript is a resubmission of an earlier submission. The following is a list of the peer review reports and author responses from that submission.
Round 1
Reviewer 1 Report
This manuscript deals with the manual annotation of the in-bed duration from thigh- and hip-worn accelerometer data using Audacity demonstrates good agreement for the time-to-bed and out-of-bed, with little mean bias and acceptable margins of agreement.
Overall the writing is sound throughout the paper. However, the novelty of the paper needs to be expanded. Here are the followings:
- It is currently only distinguished between the time in bed vs not-time in bed. However, it is important to distinguish between sleep vs awake rather than bed-time if authors focus on sleep monitoring. Please make sure to justify why it is important to this analysis when the following papers are available:
- https://www.ncbi.nlm.nih.gov/pmc/articles/PMC7680175/
- https://journals.plos.org/plosone/article?id=10.1371/journal.pone.0117382
- Throughout the paper, the author tried to strengthen the background saying that accelerometry data will be necessary for supervised machine learning. I am not convinced that this can be a strong motivation for this work. Authors can use the sleep monitoring device Zmachine® Insight+ for their gold standard. Why do we need an extra step for manual annotation of accelerometry while we have a sleep monitoring device? Rather, we can do direct machine learning in accelerometry data.
Minor:
- Please correct grammatical and spelling errors throughout the manuscript. One example is in Table 1; authors mention in the last row “Hip-Inlination” which would be I believe “Hip-Inclination”.
Reviewer 2 Report
Review for “Manual annotation of time in bed using free-living recordings of accelerometry data” by Esben Lykke, Jesper Pedersen, Niels Christian Møller, Anders Grøntved, and Jan Christian Brønd
Article summary:
This paper proposes a novel method of manually annotating time-in-bed recordings from accelerometer data using Audacity. The study is motivated by a need for a method to annotate data more easily with the aim of providing larger datasets for training supervised machine learning algorithms. The study recruited adults and children to participate in a sleep assessment for 3-4 nights (baseline) and 3 nights (follow-up) using a Zmachine Insight+ model DT-200 sensor as a ground truth metric and two accelerometers (one on the hip and one on the thigh). Each experiment was labeled by two different raters (three different raters were involved overall). Statistics on accuracy, intra, and inter-rater reliability were examined.
Review:
Major concerns:
I have two major concerns with this submission:
- The raters were given no briefing on evaluating the information and all had prior experience working with accelerometer data. This makes the experiment very difficult to reproduce, and it is not feasible for inexperienced labelers to contribute. There should be guidelines and a procedure in place for the labeling process.
- The results aim to provide more data for a supervised machine learning algorithm to identify time in bed. However, it seems worth investigating whether a heuristics-based algorithm may be able to accomplish this goal. This would also provide a benchmark for future supervised machine learning approaches to be evaluated against.
Minor concerns:
The signal features used for detection in Audacity include labels for Lying and Activity (Table 1). These appear to be provided through use of the activity type classification algorithm developed by Skotte et al. Is there a strong relationship between Lying and the in-bed and out-of-bed timings? It would seem that when lying down, the acceleration and inclination values for hip and thigh would be fairly consistent, and the activity label would clearly be “other activity” as one cannot stand, move, walk, or sit while lying down. At a naïve view, it would appear that the “lying” and “time” labels would encompass most of the unique information related to this task.
In lines 205-207, it is indicated that the labels for in-bed and out-of-bed timestamps from the accelerometers were synchronized with the ZM in-bed and out-of-bed timestamps. Does this indicate that the ZM self-labels in-bed and out-of-bed? And if so, these were used as the ground truth values for evaluation?
In Line 363, the reference ([24-26]) is not properly formatted.
The discussion should include the limitation that wearing the Zmachine EEG device could cause participants to behave in ways that are different from free-living.
Were participants allowed to take naps? Were naps considered or included in this study?
Closing parentheses are missing for some data presented in tables 3 and 4.
When manually labeling “round 2”, were labelers aware of the labels generated in “round 1” – presumably labeled by a different rater?
From Figure 3, it appears that raters became more accurate during round 2. Is there a reason why this may be the case?